# Urine NMR Metabolomics for Precision Oncology in Colorectal Cancer

**DOI:** 10.3390/ijms231911171

**Published:** 2022-09-22

**Authors:** Jesús Brezmes, Maria Llambrich, Raquel Cumeras, Josep Gumà

**Affiliations:** 1Metabolomics Interdisciplinary Group, Institut d’Investigació Sanitària Pere Virgili (IISPV), Universitat Rovira i Virgili (URV), 43204 Reus, Spain; 2Department of Electrical Electronic Engineering and Automation, Universitat Rovira i Virgili (URV), Institut d’Investigació Sanitària Pere Virgili (IISPV), 43007 Tarragona, Spain; 3Oncology Department, Hospital Universitari Sant Joan de Reus, Institut d’Investigació Sanitària Pere Virgili (IISPV), Universitat Rovira i Virgili (URV), 43204 Reus, Spain

**Keywords:** colorectal cancer, metabolomics, NMR, urine, pyruvic acid, acetic acid

## Abstract

Metabolomics is a fundamental approach to discovering novel biomarkers and their potential use for precision medicine. When applied for population screening, NMR-based metabolomics can become a powerful clinical tool in precision oncology. Urine tests can be more widely accepted due to their intrinsic non-invasiveness. Our review provides the first exhaustive evaluation of NMR metabolomics for the determination of colorectal cancer (CRC) in urine. A specific search in PubMed, Web of Science, and Scopus was performed, and 10 studies met the required criteria. There were no restrictions on the query for study type, leading to not only colorectal cancer samples versus control comparisons, but also prospective studies of surgical effects. With this review, all compounds in the included studies were merged into a database. In doing so, we identified up to 100 compounds in urine samples, and 11 were found in at least three articles. Results were analyzed in three groups: case (CRC and adenomas)/control, pre-/post-surgery, and combining both groups. When combining the case-control and the pre-/post-surgery groups, up to twelve compounds were found to be relevant. Seven down-regulated metabolites in CRC were identified, creatinine, 4-hydroxybenzoic acid, acetone, carnitine, d-glucose, hippuric acid, l-lysine, l-threonine, and pyruvic acid, and three up-regulated compounds in CRC were identified, acetic acid, phenylacetylglutamine, and urea. The pathways and enrichment analysis returned only two pathways significantly expressed: the pyruvate metabolism and the glycolysis/gluconeogenesis pathway. In both cases, only the pyruvic acid (down-regulated in urine of CRC patients, with cancer cell proliferation effect in the tissue) and acetic acid (up-regulated in urine of CRC patients, with chemoprotective effect) were present.

## 1. Introduction

Colorectal cancer (CRC) is the second leading cause of cancer death in men after lung cancer and the third leading cause of cancer death in women after breast and lung cancer [1]. To mitigate the rising burden of early-onset colorectal cancer, the American Cancer Society lowered the recommended age for screening initiation for individuals at average risk from 50 to 45 years in 2018 [2], and in 2021, the US Preventive Services Task Force concurred in a recommendation statement [3]. CRC is considered to be caused by a combination of genetic and environmental factors, where dietary factors modify the risk of colorectal adenomatous polyps, the premalignant lesion of CRC, which acquire new genetic mutations over time until cancer develops. Regarding the modifiable risk factors, the consumption of fiber, fruit, and vegetables, as well as dairy products and micronutrients such as folates and calcium, are protective against this type of cancer. In contrast, red and processed meat consumption increases the risk [4,5]. Another risk factor is obesity, and exercise and physical activity act as protectors [6]. The most frequent symptoms of colorectal cancer are changes in bowel habits and the appearance of blood in the stool. The primary diagnostic test for screening for colorectal cancer in asymptomatic patients is the detection of occult blood in feces (FOBT). This test is recommended for healthy individuals between 45/50 and 70 years of age, with biennial periodicity, and it can reduce a 25% CRC mortality [7]. FOBT can be implemented by guaiac or fecal immunochemical testing (FIT). While it has been reported that FIT outperforms guaiac [8], FIT shows considerable variability in its sensitivity and positive predictive value (PPV). The performance of these tests depends on several technical details, such as the hemoglobin threshold, the specific kit used, and the number of samples collected from each patient. Factors affecting the PPV are patient age (lower PPVs observed in younger patients), colorectal condition (lower for patients with a previous CRC clinical history), and sex (lower for females) [9]. FIT PPV values are in the range of 10–30%, while AUC (area under the ROC curve) is in the range of 0.7–0.9 [10,11]. However, even with the improvements in the performance of fecal tests, the number of false positives (FP) largely exceeds the number of true positives (TP). Improvements in CRC screening are needed to decrease the cost and potential complications of subsequent colonoscopies.

The metabolome is the complete set of small molecules (the so-called metabolites) found in a biological matrix. For this study, we focused on metabolites present in urine. Urine is obtained non-invasively, is available in larger quantities than blood, and is mostly free from interfering proteins or lipids. Moreover, the adherence to a screening program is much higher than FOBT, due to its simplicity and higher acceptance. However, urine is chemically complex as it contains the waste breakdown product of foods and beverages, environmental contaminants, drugs, endogenous waste metabolites, and bacterial byproducts [12]. Currently, ~3100 small molecules have been identified in human urine (https://urinemetabolome.ca/, accessed on 22 August 2022). Another problem with urine is the normalization process.

Nuclear Magnetic Resonance spectroscopy (NMR) and mass spectrometry are the two most widely used analytical platforms for metabolomics. Although these technologies each have their advantages and disadvantages, they may be used to complement each other [13,14]. The best advantage of NMR is its accurate absolute reproducible quantification, which is needed not only in clinical settings, but requested for in eventual clinical test adoption and regulatory approval [15]. Not only that, but NMR is perfect for quantification due to its linearity. NMR metabolomics is usually performed using an untargeted high throughput approach of the whole spectrum, which provides a complete picture of all metabolites present and quantifiable in the sample above the NMR detection limit (concentrations > 1 μM) [13,15,16]. To date, NMR metabolomics is increasingly used to successfully stratify patients [17], as well as several physiological and pathophysiological conditions [15,18].

Precision medicine, also called personalized medicine, is a prevention and treatment strategy tailored to an individual, compared to the classical approach based on the principle of “one-size-fits-all,” which does not allow more advanced treatments nowadays [19]. Therefore, an individual’s biochemical or metabolic characterization using omics technologies (genetics, epigenetics, transcriptomics, proteomics, and metabolomics) is key to precisely defining phenotypes to apply specific treatments. Since biomarkers can be considered the keystone of individualized treatment and precision medicine, metabolomics is basic for discovering novel biomarkers potentially valuable for clinical practice and unveiling alterations of the cellular function and metabolic pathway perturbations due to a given disease for a given phenotype [18]. Precision medicine promotes emerging models of critical clinical thinking supported by innovative tools and technologies, such as digitalized human big data generation and storage, artificial intelligence (AI), and machine learning (ML) [20]. For metabolomics to become a routine in precision medicine implies the direct relationship between metabolomic results and clinical decision making, similarly to any other clinical test result, in addition to the application of robust clinical laboratory standards and protocols and the availability of metabolic profiles from reference populations, defining cutoff values and decision levels [21]. In oncology, in particular, tumor molecular profiling leads to the identification of patient-specific alterations that could inform about the optimal treatments and maximize patient survival. In addition, since early diagnosis will improve the prognosis, the discovery of sensitive cancer-related biomarkers through a personalized approach has become a priority in cancer research. If applied for population screening, NMR-based metabolomics could become a powerful clinical tool in precision oncology.

In this study, with a special focus on colorectal cancer, we reviewed the metabolomics-based biomarkers from urine samples detected with NMR. In addition, we provide a detailed list of found metabolites for all studies included. Results are analyzed based on three groups: 1—case (CRC and adenomas) vs. control; 2—pre-/post-surgery; 3—a combination of both groups. A vote-counting strategy has been followed for the three groups to determine the significant compounds. For the combining group, we performed pathways and enrichment analysis.

## 2. Results

The results were divided into five parts: (1) search results, (2) characteristics of the studies included, (3) quality assurance results of the studies, (4) meta-analysis results, and (5) pathways and enrichment analysis results.

### 2.1. Search Results

The search process is shown in Figure 1. The search returned a total of eighty-three reports from Scopus (thirty-two), Web of Science (twenty-eight), and PubMed (twenty-three). From these, up to twenty-six studies were included for title and abstract screening after deleting duplications. We then excluded fourteen studies that were not related to the study question or were reviews, conference papers, book chapters, short surveys, notes, letters, or editorials. This yielded a total of twelve studies eligible for further full-text assessment. We excluded two publications because the matrix did not fit the query (no urine); the specimens were not colorectal cancer samples as they were diet-related. The final inclusion list comprises ten papers for the review where NMR is used for CRC evaluation.

### 2.2. Characteristics of the Studies Included

For the ten studies included, we prepared comprehensive tables divided on the methodology of the study (Table 1), cohort information (Table 2), and identified compounds (Appendix A). The ten reports included comprise three CRC vs. control studies [22,23,24] (one also included adenomas and hyperplastic polyps [24]), three pre-surgery/post-surgery studies [25,26,27] (one including controls [25]), three studies of adenoma samples [28,29,30], and studying the cachectic metabolites [31] (see Table 1). The two main methodology strategies used were case (CRC and adenomas) versus control analysis (seven studies, as Li Z et al., 2019 [25] also included the study of pre-surgery vs. controls), followed by the evaluation of samples before and after tumor extraction (three studies). Apart from NMR, four studies used other metabolomics platforms (mainly gas chromatography GC-MS and in one case, liquid chromatography LC-MS) to perform untargeted research. However, only two reports [28,29] validated externally the results obtained [28,29]. More commonly, internal validation was performed, usually by dividing the cohorts by training and test groups. However, validation was disclosed in less than 50% of the studies. Urine collection also differed between studies—only two studies collected first-morning urine after fasting to avoid interferences from food or lifestyle in the samples. Additionally, in two cases, spot urine was used, or information was not disclosed about the methodology followed. The method validation was performed in only 40% of the studies included. Reporting of identified and significant compounds was lacking in four of the studies [26,28,29,30]. Only one study reported both p-values and fold-changes [27].

The information of each cohort is also summarized (Table 2). Considering that complete information descriptions of participants should include age and stages of cancer, such information was only complete in six studies. Additional information (Appendix A) about body mass index (BMI) or smoking history was present in five reports, none reporting both types of information. In total, researchers from four countries have studied compounds from urine, and all these countries have a high CRC incidence and mortality rate (Appendix A). All of them have a screening program in place. Currently, only forty countries worldwide have a running screening program [32]. Three countries contribute each with three studies: Canada, China, and Germany, while the Republic of Korea (South Korea) only contributed with a single study. The number of participants per study varies from 52 [31] to 988 [24]. Only one study enrolled fewer than 100 participants, and four studies included more than 500 participants (Appendix A) [31].

### 2.3. Quality Assurance of Studies Included

Quality assurance of the studies included was performed, including ten variables for evaluation. The quality assurance results are shown in Figure 2 and Appendix A. Variables were based on the experimental methodology. The most reported domains were in sample collection, sample preparation, and experimental conditions, with more than 50% of studies reporting complete information. On the other hand, the least reported domains were in study design, statistical analysis, and analytical validation, where less than half of the studies disclosed some information.

### 2.4. Meta-Analysis Results

The total number of compounds identified in the 10 studies included was 100. Each reported compound name was translated to InChIKey with the chemical translation service [33]. These results were compared to match the compound identifiers between articles, as chemical name reporting is not usually the same. If a compound was not found by the CTS service, a manual search at PubChem (https://pubchem.ncbi.nlm.nih.gov/, accessed on 18 August 2022) was performed. In Appendix A, we provide a detailed list of all 100 compounds with their common names, molecular weight (MW), chemical formula, and major identifiers (InChIKey, PubChem ID, HMDB ID, KEGG ID, Canonical SMILES, and CAS). Appendix A presents information on the behavior of the identified compounds in the studies from the systematic search. A repeated trend means that the compound was found in more than one comparison. The molecular weight from all compounds ranged from 31.06 g/mol for methylamine with only one carbon to 408.57 g/mol for cholic acid with twenty-four carbons. From the 100 compounds, 98 are included in the human metabolome database ID (HMDB) and 88 in the Kyoto Encyclopedia of Genes and Genomes ID (KEGG).

Eleven compounds are the most repeated in the literature (three times each): acetone [24,25,31]; carnitine [25,27,28]; creatinine [22,23,25]; l-alanine [22,23,25]; l-isoleucine [25,26,27]; l-tyrosine [24,25,27]; succinic acid [25,27,28]; trans-aconitic acid [23,25,27]; trigonelline [23,24,25]; uracil [23,25,31]; urea [22,23,27]. From the 100 metabolites related to colorectal cancer identified, 76 compounds were reported just once. Compound identification was performed in seven studies, and trends in compound levels were disclosed in six of them.

Meta-analysis results followed a vote-counting strategy. Vote counting consists of the sum of the trends reported for compounds, assigning a value of +1 if the compound behavior is up-regulated, −1 if it is down-regulated, or 0 if it is equal to the comparison group. Any compound intended to be a CRC biomarker needs to be robust, meaning that it needs to be identified in more than one study, and these identifications need to all show the same trend.

#### 2.4.1. CRC and Advanced Adenoma vs. Control

Up to forty-six compounds were found to be significantly different between CRC patients or patients with advanced adenoma compared to healthy controls. Three of the included studies did not report the compound fold-change [23,28,29]; therefore, only forty of the compounds were included in the analysis (see Appendix A). Of these compounds, four were reported in two different cohorts (see Table 3). We identified only two compounds with stable behavior: creatinine and hippuric acid (both down-regulated).

#### 2.4.2. Pre-Surgery vs. Post-Surgery

There were forty compounds found to be significantly different between CRC patients pre-surgery and post-surgery (see Appendix A). Of these compounds, six were reported in two different cohorts (see Table 3). We identified four compounds with stable behavior: carnitine, d-glucose, l-lysine, and pyruvic acid (down-regulated).

#### 2.4.3. Combining Case-Control and Pre-/Post-Surgery

We considered the second group (pre- vs. post-surgery) an analog of the CRC vs. control, as pre-surgery means the patient has CRC, and post-surgery means that the patient is CRC-free. By doing so, there were seventy-four compounds found to be significantly different between case (CRC and advanced adenoma patients) and healthy controls, or by CRC pre-surgery and post-surgery (see Appendix A). Of these compounds, fourteen were reported in more than two different cohorts (see Table 3). Some of the Li et al. [25] compounds are reported in the two groups (case-control and pre-/post-surgery), as in their study they did analyze pre-surgery (CRC) vs. controls and pre-surgery vs. post-surgery, so we considered the results for each of the analyses. Vote-counting results are shown in Figure 3. The most repeated compound with the same behavior was creatinine (down-regulated). Three other compounds were reported three times (l-alanine, succinic acid, and trans-aconitic acid) but with different behaviors reported. We identified ten compounds with stable behavior: creatinine, 4-hydroxybenzoic acid, acetone, carnitine, hippuric acid, l-threonine, pyruvic acid (down-regulated), acetic acid, phenylacetylglutamine, and urea (up-regulated).

### 2.5. Pathways and Enrichment Analysis

The significant compounds found in the combination analysis (case-control and pre-/post-surgery), are shown in Table 4 along with the relevant identifiers. The pathways and enrichment analysis of these compounds showed 15 pathways (Figure 4A). The most relevant pathways (Figure 4B) are the pyruvate metabolism (*p*-value = 0.006) and the glycolysis/gluconeogenesis pathway (*p*-value = 0.009). In both pathways, pyruvic acid (down-regulated) and acetic acid (up-regulated) are included.

## 3. Limitations of This Work

Several confounding elements might affect the obtained results in this review. Some NMR-based metabolomics of CRC profile in urine samples sometimes consider heterogeneous groups of cases (e.g., patients with different cancer stages, patients with also other cancer types, etc.). If not properly considered, these factors represent important confounding elements masking biological results. For that reason, we grouped the results for those more homogenous groups, also aiming to increase the number of studies to be compared. Another confounding element is the gender effect [36]. We have not been able to evaluate it, as the included studies do not provide metabolites behaviors in this regard, nor analysis of covariance (ANCOVA). However, from the list provided by Fan et al. of compounds altered in urine by gender effect [36], none of them are found as significant in any of the relevant compounds in the colorectal cancer analysis (Table 3) for any of the studied groups. Indeed, urine metabolomic analysis could be easily implemented to be used as wide-scale population screening. However, in clinics, the biggest drawback of urine metabolomics’ profile is the samples’ variability due to lifestyle, diet, environmental factors, and the pathophysiological status of the patients. We have tried to account for some of these factors by including the country of the study’s origin (Appendix A). From the ten included studies, only five have compounds found as significant in the meta-analysis. Of those, three are Asiatic-diet (two China; one Republic of Korea), and two are Western-diet (Canada and Germany). However, no significant result from diet-specific could be achieved given the limited number of studies doing NMR for CRC urine evaluation. Nevertheless, six significant compounds of the meta-analysis (acetone, carnitine, creatinine, l-threonine, pyruvic acid, and urea) are detected in the same trend with participants of both regions, Asiatic and Western. Finally, given the small number of studies included, the conclusions of this review might change in future meta-analyses; therefore, readers should use caution when using the results of this review.

Regarding the technology, low sensitivity has always been the primary limitation of NMR spectroscopy. Although significant signal enhancement using cryo-probes, higher magnetic fields, and digital signal processing has improved the NMR sensitivity, many important, low-abundance metabolites still cannot be detected with today’s NMR technology. It is widely acknowledged that there are several thousand measurable or detectable metabolites in human urine, but from those, only a few hundred metabolites (the most abundant) have been reported as being reliably detected by NMR [12,37]. While high-abundance metabolites are almost always physiologically important, low-concentration metabolites are often more important as diagnostic biomarkers. This means that NMR-based metabolomics is often unable to detect these important molecules.

## 4. Discussion

One of the biggest efforts in this review was the merging and curation of all relevant compounds from the selected articles. Each individual compound from each article was searched for its InChIKey. To harmonize compounds names, we selected them from the PubChem ID associated with the InChIKey at PubChem. We have also included several chemical identifications (Appendix A), so it will be easier to compare the results presented here with future results reported by the scientific community.

For the systematic review, one hundred compounds were identified in urine samples among individuals that participated in the studies with colorectal cancer or adenomas, but only twenty-five were reported more than once. Of those, the most abundant compounds were carboxylic acids and derivatives, comprising fifteen compounds (including ten amino acids and derivatives, three dicarboxylic and tricarboxylic acids and their derivatives, one alpha-keto acid and derivative, and one urea), four organoheterocyclic compounds (two indoles, one furanone, and one diazine), two organic nitrogen compounds (carnitines and cholines), one benzenoid (benzamidas), one alkaloid and derivative, and one organic oxygen compound (ketone).

The studies included were divided into three groups, and the analysis of significant compounds was conducted via vote-counting. (1) CRC and advanced adenoma vs. control, which returned two significant down-regulated compounds (creatinine and hippuric acid); (2) pre-surgery vs. post-surgery patients, which returned also two significant down-regulated compounds (carnitine and pyruvic acid); (3) a combination of groups one and two, which returned twelve significant compounds, including the four significant compounds from groups one and two, but with eight new significant compounds. From the significant compounds, nine are down-regulated (creatinine*, 4-hydroxybenzoic acid, acetone, carnitine*, d-glucose, hippuric acid*, l-Lysine, l-threonine, and pyruvic acid*) and three are up-regulated (acetic acid, phenylacetylglutamine, and urea). Compounds with an asterisk indicate those found in groups one and two. For group three, we conducted a pathway and enrichment analysis, and two pathways were found to be significant: pyruvate metabolism and glycolysis and gluconeogenesis.

In many metabolomics protocols of urine, a common practice is to normalize the volume of samples with the concentration of creatinine. Due to this normalization, no possible significant alteration in creatinine will be observed. In our case, only two studies reported normalizing with creatinine [27,31], and none reported creatinine as a significant compound. Hippuric acid appears at abnormal levels in urine in conditions related to hepatic function, renal system, and metabolic disorders. Goveia et al. [38] evaluated 12 kinds of cancers in 25 studies and hippuric acid was the only common significant compound in urine. Additionally, Mallafré-Muro et al. [39] found hippuric acid and pyruvic acid compounds significantly altered in colorectal cancer, both down-regulated as we have reported. However, hippuric acid is the urinary metabolite most strongly related to fecal microbial richness [40], is commonly altered in almost all malignancies and a wide variety of other diseases [41], and has also been reported as an up-regulated marker of fruit and vegetable intake [42]. D-glucose should be treated with caution due to its relation to diabetes onset in urine. The studies included do not report co-morbidities; therefore, we cannot know if the relevant d-glucose metabolite is due to another metabolic disorder. L-lysine is an essential amino acid that is found in great quantities in muscle tissues and stimulates calcium absorption, carnitine synthesis, and growth and repair of muscle tissue. L-lysine has been associated with diabetes and cardiovascular diseases [43]. Carnitine is an amino acid derivative that has many metabolic functions, including stimulating hematopoiesis, preventing programmed cell death in immune cells, inhibiting collagen-induced platelet aggregation, and modulating fatty acid oxidation. Carnitine palmitoyltransferase I (CPTI) was reported to be overexpressed in numerous tumors, suggesting it may play an important role in tumor neovascularization [44]. Therefore, the carnitine system is a pivotal mediator in cancer metabolic plasticity, intertwining key pathways, factors, and metabolites that supply an energetic and biosynthetic demand for cancer cells [45]. Serine racemase (SRR) supports proliferation of colorectal cancer cells by the dehydration of l-serine and d-serine, resulting in the formation of pyruvate and ammonia [46]. SRR contributes to the pyruvate pool in colon cancer cells, enhances proliferation, maintains mitochondrial mass, and increases basal reactive oxygen species production, which has anti-apoptotic effects. As neoplastic cells fuel with pyruvate, its amount is decreased in urine.

A combination of both groups was performed with the premise that CRC post-surgery could be an analog of a healthy condition; however, some patients after surgery have micro metastasis not detectable at the time of surgery; therefore, strictly, a few patients would not be cancer-free patients. When combining both groups, up to 12 compounds were found relevant. From those, creatinine was the most found in the analyzed studied. The mentioned significant compounds per group are also relevant in the combined analysis (creatinine, hippuric acid, carnitine, d-glucose, l-lysine, and pyruvic acid all down-regulated), along with 4-hydroxybenzoic acid, acetone, l-threonine (down-regulated) and acetic acid, phenylacetylglutamine, and urea (up-regulated).

The 4-hydroxybenzoate, creatinine, and acetate had significantly different metabolite levels among bladder cancer, prostate cancer, and renal cell carcinoma [47], and 4-hydroxybenzoic acid was found elevated in the urine of gastric cancer patients [48]. Acetone as a urinary volatile has been reported to discriminate colorectal cancer patients from healthy controls [49]. The l-threonine amino acid is vital for human health, but it cannot be synthesized by the human body and, therefore, must be obtained from a diet. Moreover, it has been associated significantly in the urine of ovarian cancer patients [50]. Acetic acid, on the contrary, has an apoptotic effect [51] and for that reason, its value is up-regulated in the urine. Phenylalanine is ingested via the consumption of food. Some phenylalanine reaches the large intestine and is metabolized by the intestinal flora to form phenylacetic acid, which is then transported to the liver by the circulatory system to combine with glutamine, ultimately resulting in the production of phenylacetylglycine (a major metabolite in mice) and phenylacetylglutamine (a major metabolite in humans) [35]. Phenylacetylglycine may sometimes be mistaken for phenylacetylglutamine in NMR measurements [34], and for that reason, we changed the original identification of phenylacetylglycine reported by Li et al. [25] to phenylacetylglutamine. In fact, Mallafré-Muró et al. report in a meta-analysis a list of 244 compounds found in the urine of colorectal cancer, both liquid and gas phases, and only phenylacetylglutamine is reported [39]. As an interesting note, from the 100 compounds listed from the included studies, we have 17 new compounds not previously reported in the 244 compounds list [39]. Finally, urea is the most abundant metabolite in urine, and several studies report the utilization of urease enzymes to remove it from the samples. Further evaluation of urea must be taken cautiously.

The pathways and enrichment analysis returned only two pathways significantly expressed: the pyruvate metabolism and the glycolysis/gluconeogenesis pathway. In both cases, only the pyruvic acid (down-regulated) and acetic acid (up-regulated) were included. We can conclude that those two compounds have an opposing effect of enhancing cancer cell proliferation (pyruvic acid) and a chemoprotective effect (acetic acid).

This review aimed to highlight relevant results obtained for colorectal cancer diagnosis using metabolomics by NMR and the possible role of this approach in the clinical practice. NMR-based metabolomics is a fast, high-throughput, robust, and reproducible technique [15]; thus, moving from the analysis of hundreds to thousands of samples is realistically an approachable target [52]. This review on NMR was conducted for the low requirements of sample preparation and for the quantitative behavior of the technique, which does not require true standards and calibration curves, making easier its translation to clinics. Therefore, NMR metabolomics is an essential component in precision medicine as well as biomarker discovery and its translation to personalized clinical care strategies.

## 5. Materials and Methods

### 5.1. Search Sentence (Query)

The search sentence used was TITLE-ABS-KEY ((urine OR urinary OR urinate OR urination) AND (colorectal OR colon) AND (tumor OR tumour OR malignancy OR neoplasm OR cancer OR carcinoma OR adenoma OR polyps OR polyp) AND (human OR humans) AND (NMR or {nuclear magnetic resonance}) AND ({metabolite profiling} OR {metabolite analysis} OR {metabolic profiling} OR {metabolic fingerprinting} OR {metabolic characterization} OR metabolome OR metabolomics OR metabolomic OR metabonomics OR metabonomic OR lipidome OR lipidomics OR lipidomic)).

This sentence was searched in PubMed (https://pubmed.ncbi.nlm.nih.gov/), Web of Science (WOS) (https://webofknowledge.com/), and SCOPUS (https://www.scopus.com/) on 30 July 2022.

### 5.2. Inclusion and Exclusion Criteria

Data on title, year of publication, authors, and abstracts were combined in an Excel file for each searcher engine used. In the initial screening steps, duplicated articles among databases were removed, and then, reviews, book chapters, conference papers, etc., were excluded. This initial screening process was employed by reading the titles and abstracts of the articles. In the eligibility step, articles were further evaluated by reading their full texts. Eligibility was reviewed by at least two authors to avoid personal biases, and a decision was made by consensus when there were inconsistencies. We excluded any studies if (1) the matrix did not fit the query (no urine); (2) the studies were conducted on animals or cell lines; (3) the study was not on colorectal cancer; (4) the study was related to food or drug outcomes. There was no restriction to the study design, race, geographical area, or certain population for the systematic review.

### 5.3. NMR CRC Database Creation

In Appendix A, we provide a detailed list of all 100 compounds with their common names, MW, chemical formula, and major identifiers (InChIKey, PubChem ID, HMDB ID, KEGG ID, SMILES, CAS number) and chemical classification. The chemical translator service [33] (CTS, http://cts.fiehnlab.ucdavis.edu/, accessed on 15 August 2022) was used to retrieve the chemical compound information and identifiers, while any missing information was checked at PubChem (https://pubchem.ncbi.nlm.nih.gov/, accessed on 16 August 2022). Compounds’ chemical classes were retrieved using ClassyFire [53] (http://classyfire.wishartlab.com/, accessed on 18 August 2022).

### 5.4. Statistical Analysis

The vote-counting plot was produced with the Amanida R-package [54] (https://ubidi.shinyapps.io/easy-amanida/, accessed on 22 August 2022). Pathways and enrichment analysis were performed with MetaboAnalyst 5.0 [55] (https://www.metaboanalyst.ca/, accessed on 24 August 2022). Finally, the incidence vs. mortality plot by country was plotted using the open-source data visualization framework RAWGraphics [56] (https://app.rawgraphs.io/, accessed on 24 August 2022).

## Figures and Tables

**Figure 1 ijms-23-11171-f001:**
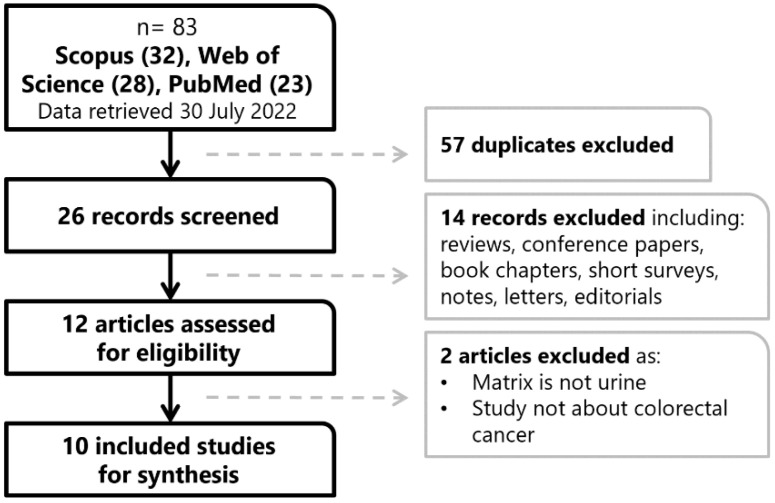
Workflow of the review search process.

**Figure 2 ijms-23-11171-f002:**
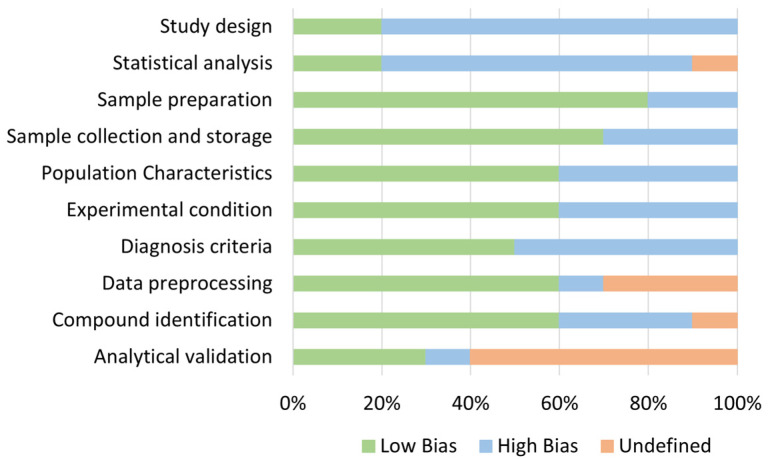
Quality assessment results for the included studies.

**Figure 3 ijms-23-11171-f003:**
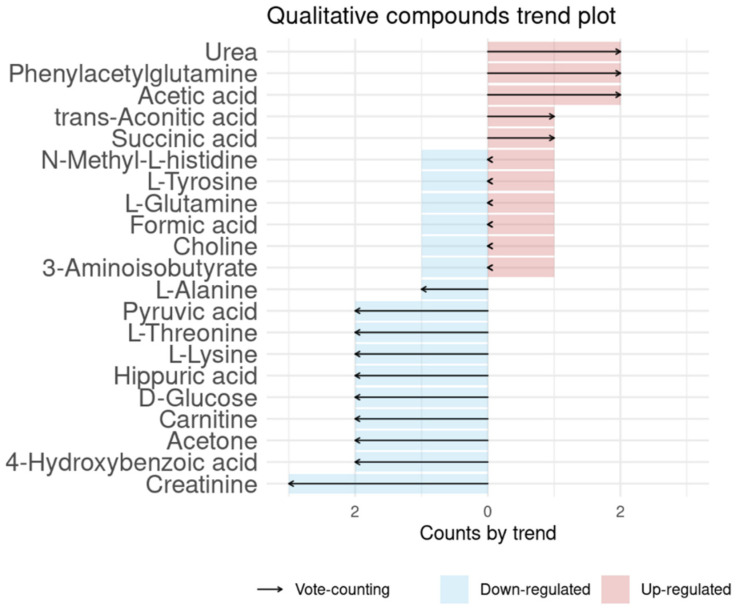
Vote-counting of colorectal cancer-related compounds. Red positive values are compounds up-regulated in CRC, while blue negative values are compounds down-regulated in CRC.

**Figure 4 ijms-23-11171-f004:**
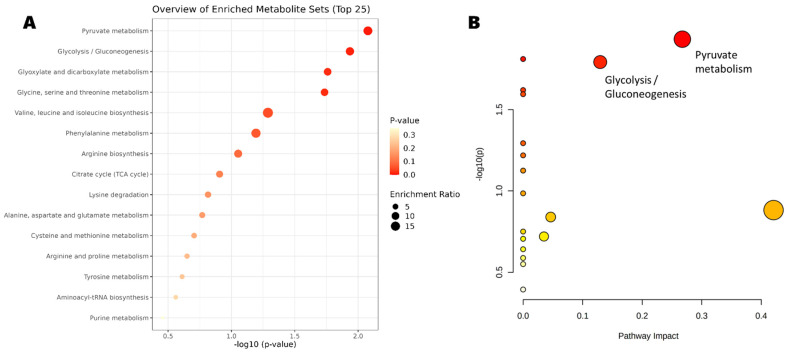
(**A**) Enrichment and (**B**) pathways analyses.The x-axis represents the pathway impact value computed from pathway topological analysis, and the y-axis is the-log of the *p*-value obtained from pathway enrichment analysis. The pathways that were most significantly changed are characterized by both a high-log(p) value and high impact value (top right region). The node color is based on its *p*-value and the node radius is determined based on their pathway impact values.

**Table 1 ijms-23-11171-t001:** Methodological information from the studies included.

Ref.	Platform	Type of Study	Ethics Approval	Urine Collection	Urine Storage	Analytical Validation	ROC Curve (Training/Testing)
[22]	NMR	CRC/control	yes	First-morning urine	−80 °C	-	0.823 taurine, 0.783 alanine, 0.842 3-aminoisobutyrate/ND
[23]	1H-NMR	CRC/control (including stages + other cancer types)	yes	Fasting morning urine	−80 °C	80% training, 20% testing	0.875 alanine, 0.913 glutamine, 0.933 aspartic acid/ND
[24]	1H-NMR	Positive colonoscopy (adenomas, hyperplastic, CRC)/control	yes	Midstream urine	4 h at 4 °C 24 h at −80 °C	27-fold cross-validation	0.715 (4 compounds)/ND
[25]	1H-NMR	CRC pre-/post-surgery and post-chemotherapy	yes	Morning urine	−80 °C	-	-
[26]	1H-NMR + GC-MS	CRC pre-surgery and post-surgery (6/12 months)	yes	Urine spot	−80 °C	-	-
[27]	1H-NMR + GC-MS	CRC pre-/post-surgery and 6-/12-month follow-up AND intra-stages	yes	Pre-/post-surgery overnight fasting urine, 6-/12-month follow-up URINE spot	−80 °C	-	0.89 (20 compounds)/ND
[28]	NMR + targeted LC-MS/MS	Adenoma/control	yes, with ID	Midstream urine	−80 °C ^‡^	2/3–1/3	0.687/0.692
[29]	1D NMR	Adenoma/control	yes	Midstream urine	4 h at 4 °C 24 h at −80 °C	Validation of [28]	0.717/ND
[30]	1D NMR	Adenoma/control	yes	Midstream urine	4 h at 4 °C 24 h at −80 °C	2/3–1/3	0.752/ND
[31]	1H-NMR + GC-MS	CRC cachectic/pre-cachectic /non-cachectic	yes, with ID	ND	−80 °C	-	-

ND: not disclosed. MS/MS: tandem mass spectrometry, LC: liquid chromatography. ^‡^ Urine temperature conditions were reported in a previous publication.

**Table 2 ijms-23-11171-t002:** Cohort information of selected studies.

Ref.	Group	N	Age (Error and Type)	Male/Female	Cancer Staging Classification (n)	Country
[22]	CRC	92	60 (R: 32–85)	62/30	0 (24), I (8), II (7), III (13), IV (4)	KR
	Control	156	52 (R: 22–76)	76/80	-	
[23]	CRC	55	60 (ND)	26/29	I/II (23), III/IV (32)	CN
	Control	40	59 (ND)	19/21	-	
	EC	18	61 (ND)	8/10	-	
[24]	Colonoscopy (CRC)	2	ND	ND	ND	CA
	Colonoscopy (adenoma)	243	ND	ND	-	
	Colonoscopy (hyperplastic)	110	ND	ND	-	
	Colonoscopy (all)	355	58.9 (SD: 8.2) ‡	196/159	ND	
	Control	633	56.2 (SD: 8.1) ‡	269/364	-	
[25]	CRC pre-S	25	56.5 (SD: 14.1)	18/7	II (8), III (17)	CN
	CRC post-S	25	58.5 (SD: 12.9)	18/7	II (11), III (14)	
	CRC post-C	25	52.3 (SD: 13.7)	16/9	II (6), III (19)	
	Control	31	52.3 (SD: 11.4)	21/10	-	
[26]	CRC pre-S	163	64 (SD: 12)	110/53	I/II (76), III/IV (87)	DE
	CRC post-S (6 m)	83	62 (SD: 12)	60/23	I/II (36), III/IV (47)	
	CRC post-S (12 m)	57	61 (SD: 10)	39/18	I/II (32), III/IV (25)	
[27]	CRC pre-S	97	64.8 (SD: 12.9)	59/38	0 (5), I (12), II (40), III (22), IV (18)	DE
	CRC post-S	12	63.9 (SD: 12.5)	10/2	0 (0), I (4), II (4), III (2), IV (2)	
	CRC (6 m)	52	60.1 (SD: 11)	38/14	0 (0), I (12), II (17), III (15), IV (8)	
	CRC (12 m)	38	61.5 (SD: 11.6)	24/14	0 (0), I (7), II (13), III (14), IV (4)	
[28]	Adenoma	155	59.9 (SD: 7.4)	95/60	ND	CA
	Control	530	56.1 (SD: 8.2)	222/308	-	
[29]	Adenoma	345	65.1 (SEM: 6.6)	197/148	ND	CN
	Control	316	61.8 (SEM: 7.4)	82/234	-	
[30]	Adenoma	243	59.5 (SEM: 0.67)	145/98	ND	CA
	Control	633	55.8 (SEM: 0.47)	269/364	-	
[31]	CRC Cac	16	58.38 (ND: 10.33)	11/5	I (5), II (1), III (6), IV (4)	DE
	CRC pre-Cac	13	55.84 (ND: 11.67)	11/2	I (2), II (5), III (4), IV (2)	
	CRC non-Cac	23	62.74 (ND: 12.22)	14/9	I (7), II (9), III (7), IV (0)	

ND: not disclosed, S: surgery, m: month, C: chemotherapy, Cac: Cachectic, SD: standard deviation, SEM: standard error of the mean, R: range, EC: esophageal cancer, CN: China, KR: South Korea, CA: Canada, DE: Germany. ‡ Total of participant’s data. Cancer stages follow T-stage (0, I, II, III, IV).

**Table 3 ijms-23-11171-t003:** Relevant compounds per studied group. The compounds shown are found in at least 2 different cohorts. Compounds in bold have a vote count of at least ±2.

Common Name	No. of Cohorts	Behavior (Up–Down–Equal)	Vote-Counting	N	Reference
**CRC and advanced adenoma vs. Control**					
**Creatinine**	2	0–2–0	−2	343	[22,23]
**Hippuric acid**	2	0–2–0	−2	343	[22,23]
Choline	2	1–1–0	0	151	[23,25]
L-Alanine	2	1–1–0	0	343	[22,23]
**Pre-surgery vs. Post-surgery**					
**Carnitine**	2	0–2–0	−2	185	[25,27]
**D-Glucose**	2	0–2–0	−2	112	[25] ^†^
**L-Lysine**	2	0–2–0	−2	112	[25] ^†^
**Pyruvic acid**	2	0–2–0	−2	185	[25,27]
Succinic acid	2	1–1–0	0	185	[25,27]
Trans-Aconitic acid	2	1–1–0	0	185	[25,27]
(**CRC and advanced adenoma vs. Control**) **AND** (**Pre-surgery vs. Post-surgery**)					
**Creatinine**	3	0–3–0	−3	399	[22,23,25]
**4-Hydroxybenzoic acid**	2	0–2–0	−2	112	[25] ^†^
**Acetone**	2	0–2–0	−2	1044	[24,25]
**Carnitine**	2	0–2–0	−2	191	[25,27]
**D-Glucose**	2	0–2–0	−2	112	[25] ^†^
**Hippuric acid**	2	0–2–0	−2	343	[22,23]
**L-Lysine**	2	0–2–0	−2	112	[25] ^†^
**L-Threonine**	2	0–2–0	−2	383	[22,27]
**Pyruvic acid**	2	0–2–0	−2	191	[25,27]
L-Alanine	3	1–2–0	−1	399	[22,23,25]
Choline	2	1–1–0	0	151	[23,25]
3-Aminoisobutyrate	2	1–1–0	0	304	[22,25]
Formic acid	2	1–1–0	0	112	[25] ^†^
L-Glutamine	2	1–1–0	0	151	[23,25]
L-Tyrosine	2	1–1–0	0	1123	[24,27]
N-Methyl-L-histidine	2	1–1–0	0	112	[25] ^†^
Succinic acid	3	2–1–0	1	247	[25,27] ^†^
Trans-Aconitic acid	3	2–1–0	1	286	[23,25,27] ^†^
**Acetic Acid**	2	2–0–0	2	112	[25] ^†^
**Phenylacetylglutamine ***	2	2–0–0	2	112	[25] ^†^
**Urea**	2	2–0–0	2	383	[22,27]

^†^ Reference [25] has different analyses of a cohort with different time points. * The original study reported phenylacetylglycine. If NMR-based identification is based on signals from the benzyl group, it is likely to be mistaken with phenylacetylglutamine, which contains a similar group with overlapping signals [34]. Additionally, phenylacetylglycine has not been identified in human urine [35].

**Table 4 ijms-23-11171-t004:** Significant compounds in (colorectal cancer and advanced adenoma vs. control) and (pre-surgery and post-surgery) comparisons, including the relevant identifiers. Asterisk indicates that the compound is found relevant in the pathways and enrichment analysis.

Compound Name	MW	Chemical Formula	PubChem ID	HMDB ID	KEGG ID	Reference
Creatinine	113.12	C4H7N3O	588	HMDB0000562	C00791	[22,23,25]
4-Hydroxybenzoic acid	138.12	C7H6O3	135	HMDB0000500	C00156	[25] ^†^
Acetone	58.08	C3H6O	180	HMDB0001659	C00207	[24,25]
Carnitine	161.20	C7H15NO3	288	HMDB0000062	C00318	[25,27]
D-Glucose	180.16	C6H12O6	5793	HMDB0000122	C00031	[25] ^†^
Hippuric acid	179.17	C9H9NO3	464	HMDB0000714	C01586	[22,23]
L-Lysine	146.19	C6H14N2O2	5962	HMDB0000182	C00047	[25] ^†^
L-Threonine	119.12	C4H9NO3	6288	HMDB0000167	C00188	[22,27]
* Pyruvic acid	88.06	C3H3O3	1060	HMDB0000243	C00022	[25,27]
* Acetic Acid	60.05	C2H4O2	176	HMDB0000042	C00033	[25] ^†^
Phenylacetylglutamine	264.28	C13H16N2O4	92,258	HMDB0006344	C05598	[25] ^†^
Urea	60.06	CH4N2O	1176	HMDB0000294	C00086	[22,27]

^†^ Reference [25] has different analyses of a cohort with different time points.

## Data Availability

The data presented in this study are openly available in Zenodo at [10.5281/zenodo.7044970].

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
