# Peer review of "Urine NMR Metabolomics for Precision Oncology in Colorectal Cancer"

_ijms, 2022, doi:10.3390/ijms231911171_

Round 1

Reviewer 1 Report

The manuscript by Brezmes et al. aimed to systematically review the colorectal cancer biomarkers from urine samples by studying the metabolomic profile through a NMR analysis. The manuscript is well-written, organized and really easy to read, despite the topic difficulty. However, I have some few suggestions for the Authors in order to improve the manuscript. For that reasons, I suggest minor revision of the manuscript before to assess it for publication in Internation Journal of Molecular Sciences. Particularly:

- It will be useful to insert a flow diagram to better explain study selection;

- Page 6, line 172: Please replace “Table S2” with “Table S3”.

Author Response

Comments and Suggestions for Authors
The manuscript by Brezmes et al. aimed to systematically review the colorectal cancer
biomarkers from urine samples by studying the metabolomic profile through a NMR analysis.
The manuscript is well-written, organized and really easy to read, despite the topic difficulty.
However, I have some few suggestions for the Authors in order to improve the manuscript. For
that reasons, I suggest minor revision of the manuscript before to assess it for publication in
Internation Journal of Molecular Sciences.

Thanks for considering our article for publication in International Journal of Molecular
Sciences. We are glad that you find the article interesting and suitable for minor revision.

Particularly:

- It will be useful to insert a flow diagram to better explain study selection;

We agree with the reviewer that including it will improve the readability of the review.
Therefore, we have included a workflow figure of the search process, as Figure 1. Other
captions have been re-numbered accordingly.

- Page 6, line 172: Please replace “Table S2” with “Table S3”.

Many thanks for the appreciation, we got confused with the numbers. It has been corrected.

Thanks again for your comments and time reviewing our systematic review and meta-analysis

Reviewer 2 Report

This review provides an exhaustive evaluation of NMR metabolomics for the determination of colorectal cancer in urine. It is an interesting technique that can be useful for diagnosis. I have only some minor comments:

1. In the abstract, please define CRC in the first mention and specify in what condition a metabolite is “down-regulated” or “up-regulated”.

2. It would be interesting, if possible, to see the differences between gender and race, or at least the country of the study. If not possible, please state why. 

3. Please, state the limitations of the work. Is the number of studies analyzed here big enough for reaching conclusions?

Author Response

This review provides an exhaustive evaluation of NMR metabolomics for the determination of
colorectal cancer in urine. It is an interesting technique that can be useful for diagnosis.

Thanks for considering our article for publication in International Journal of Molecular
Sciences. We are glad that you find the article interesting and suitable for minor revision.

I have only some minor comments:

1. In the abstract, please define CRC in the first mention and specify in what condition a
metabolite is “down-regulated” or “up-regulated”.

Many thanks for pointing it out. We have included “(CRC)” the first time it appears in the
abstract. Also, up-regulated and down-regulated conditions have been stated at the end of the
abstract.

2. It would be interesting, if possible, to see the differences between gender and race, or at
least the country of the study. If not possible, please state why.

We agree with the reviewer, that gender, race and even diet (country-dependent) would be
very interesting to determine if they affect to the CRC panel of metabolites. At the
supplementary tables S1 and S5-7 we have included a column of “Country/ies” so each
compound could be linked to the country of the study included in the analysis.

Even though we have tried to see differences between gender, race and diet (country-
dependent), given the limited amount of studies analyzed, this could not be properly achieved.
This issue has been raised in the new section “Limitations”, as requested in your next
suggestion.

3. Please, state the limitations of the work. Is the number of studies analyzed here big enough
for reaching conclusions?

We agree with the reviewer, a section of limitations was missing. We have included it as
section “3. Limitations of this work”. It is placed before the discussion. Sections have been re-
numbered accordingly. The new section includes not only concerns on the number of the
studies analyzed, but also about the gender, race and diet (country-dependent), as pointed in
the previous suggestion.

The new section is:

3. Limitations of this work
Several confounding elements might affect the obtained results in this review. Some NMR-
based metabolomics of CRC profile in urine samples sometimes consider heterogeneous group
of cases (e.g., patients with different cancer stages, patients with also other cancer types, etc.).
If not properly considered, these factors represent important confounding elements masking
biological results. For that reason, we grouped the results for those more homogenous groups,
also aiming to increase the number of studies to be compared. Another confounding element
is the gender effect [36]. We have not been able to evaluate it, as the included studies do not
provide metabolites behaviors in this regard, nor analysis of covariance (ANCOVA). However,
from the list provided by Fan et al. of compounds altered in urine by gender effect [36], none
of them is found as significant in any of the relevant compounds in the colorectal cancer
analysis (Table 3), for any of the studied groups. Indeed, urine metabolomic analysis could be
easily implemented to be used as wide-scale population screening. However, in clinics, the
biggest drawback of urine metabolomics’ profile is the samples' variability due to lifestyle, diet,
environmental factors, and the pathophysiological status of the patients. We have tried to
account for some of these factors by including the country of the study’s origin (Tables S1 and
S5-7). From the 10 included studies, only 5 have compounds found as significant in the meta-
analysis. Of those, three are Asiatic-diet (2 China, 1 Republic of Ko-rea), and two are Western-
diet (Canada, Germany). However, no significant result from diet-specific could be a
chieved,
given the limited number of studies doing NMR for CRC urine evaluation. Nevertheless, 6

significant compounds of the meta-analysis (acetone, carnitine, creatinine, l-threonine, pyruvic
acid, and urea) are detected in the same trend with participants of both regions, Asiatic and
Western. Finally, given the small number of studies included, the conclusions of this review
might change in future meta-analyses, and therefore, readers should use caution when using
the results of this review.

Regarding the technology, low sensitivity has always been the primary limitation of NMR
spectroscopy. Although significant signal enhancement using cryo-probes, higher magnetic
fields, and digital signal processing has improved the NMR sensitivity, many important, low-
abundance metabolites still cannot be detected with today’s NMR technology. It is widely
acknowledged that there are several thousand measurable or detectable metabolites in
human urine, but from those, only a few hundred metabolites (the most abundant) have been
reported as being reliably detected by NMR [12,37]. While high-abundance metabolites are
almost always physiologically important, low-concentration metabolites are often more
important as diagnostic biomarkers. This means that NMR-based metabolomics is often unable
to detect these important molecules.

References from the section:
New references 36 and 37 have been included. Other references have been updated
accordingly.

12. Bouatra, S.; Aziat, F.; Mandal, R.; Guo, A.C.; Wilson, M.R.; Knox, C.; Bjorndahl, T.C.; Krishnamurthy,
R.; Saleem, F.; Liu, P.; et al. The Human Urine Metabolome. PLoS One 2013, 8, e73076,
doi:10.1371/journal.pone.0073076.

36. Fan, S.; Yeon, A.; Shahid, M.; Anger, J.T.; Eilber, K.S.; Fiehn, O.; Kim, J. Sex-Associated Differences in
Baseline Urinary Metabolites of Healthy Adults. Sci Rep 2018, 8, 11883, doi:10.1038/s41598-018-29592-
3.

37. Wishart, D.S.; Guo, A.; Oler, E.; Wang, F.; Anjum, A.; Peters, H.; Dizon, R.; Sayeeda, Z.; Tian, S.; Lee,
B.L.; et al. HMDB 5.0: The Human Metabolome Database for 2022. Nucleic Acids Res 2022, 50, D622
D631, doi:10.1093/nar/gkab1062.

Thanks again for your comments and time reviewing our systematic review and meta-analysis.
We have tried to answer the best that we could all your concerns and made the proper
changes to the manuscript.